# Edge Bleeding Artifact Reduction for Shape from Focus in Microscopic 3D Sensing

**DOI:** 10.3390/s23208602

**Published:** 2023-10-20

**Authors:** Sang-Ho Park, Ga-Rin Park, Kwang-Ryul Baek

**Affiliations:** School of Electric and Electronic Engineering, Pusan National University, Busan 46241, Republic of Korea; propia@pusan.ac.kr (S.-H.P.); parkhm7446@pusan.ac.kr (G.-R.P.)

**Keywords:** shape from focus, 3D sensing, depth estimation, edge bleeding artifact reduction, microstructures

## Abstract

Shape from focus enables microscopic 3D sensing by combining it with a microscope system. However, edge bleeding artifacts of estimated depth easily occur in this environment. Therefore, this study analyzed artifacts and proposed a method to reduce edge bleeding artifacts. As a result of the analysis, the artifact factors are the depth of field of the lens, object texture, brightness difference between layers, and the slope of the object. Additionally, to reduce artifacts, a weighted focus measure value method was proposed based on the asymmetry of local brightness in artifacts. The proposed reduction method was evaluated through simulation and implementation. Edge bleeding artifact reduction rates of up to 60% were shown in various focus measure operators. The proposed method can be used with postprocessing algorithms and reduces edge bleeding artifacts.

## 1. Introduction

Three-dimensional sensing computes a depth map with data obtained by sensing an object and restoring the object in a 3D image close to the real object based on the obtained depth map. Image depth estimation methods for 3D sensing include structured light methods, infrared light or laser methods, stereoscopy, and shape-from-focus (SFF). The method using structured light estimates the depth using the shape transformation of captured structured light by projecting structured light onto an object [1,2,3]. Infrared or laser light methods estimate the depth from the differences in the angle or phase of reflected light [4,5]. Stereoscopy is a method of estimating depth using the shape difference between the images captured by two cameras at different positions [6,7,8]. SFF estimates depth using the degree to which each image is in the focus of the image sequences captured with different focal distances with a single camera [9,10,11]. The focal distance means the distance between the lens and the focus position in the object plane. Unlike other methods, SFF has a simple structure because it does not require additional devices other than a camera. Furthermore, SFF can easily realize μm-scale 3D sensing by combining it with a microscope system [12,13].

Figure 1a presents a depth estimation method using SFF. First, sequence images with different focal distances in the same scene are obtained. The focal distance could be changed using the lens focus adjustment ring or by changing the distance between the lens and the object. Second, the focus measure (FM) operator is applied to obtain local focus information and measure the degree of focus of each image. The FM operators are most important in the quality of SFF results. Operators based on Laplacian, wavelet, gradient, and variance are commonly used and have different advantages and disadvantages depending on the characteristics of the sensing environment [14]. In addition, the FM value can be corrected according to the purpose. Next, the distance corresponding to the index with the largest FM value is estimated as the depth for the corresponding pixel of the object by comparing the FM value according to the focal distance in the FM value volume. The estimated depth map is noisy in low-light images or low-texture objects, so it requires postprocessing.

When blurred by defocus, components, such as contrast, gradient, variance, and high frequency of the image, are reduced. Various FM operators have been proposed to measure the degree of focus using the image sharpness based on these characteristics. Gradient-based operators calculate the first derivative or gradient of images [15,16]. This assumes that the defocused image is blurrier than the in-focused image. Laplacian-based operators compute the second derivative or Laplacian of images [17]. This operator has the same assumptions as gradient-based operators. Wavelet-based operators calculate the frequency component of images using a discrete wavelet transform [18,19]. The defocused image has fewer high-frequency components than the in-focus image. Statistics-based operators compute some statistical values, such as the variance [20,21]. This assumes that the defocused image has a smaller variance component than the in-focused image.

When SFF is performed in a microscopic environment, depth blurring occurs more often at the edges than in a general environment. This artifact is called an edge bleeding artifact, in which the depth estimated from the object edge spreads into continuous values. Figure 1b shows that depth is blurred at the edge between the wire and the background. In microscopic images, the depth of field (DoF) is very short, so defocus blurring affects a wide area [22,23]. When an object is blurred, the gradient component in the area around the edge increases, which is measured by the FM operator. Nevertheless, the FM value around the edge becomes larger than the FM value of the object if the texture is weak, which causes edge bleeding artifacts, regardless of whether the texture of the object is strong. This artifact has the characteristic that postprocessing algorithms do not easily remove it because it forms a continuous value with the depth of the surrounding area.

In general, postprocessing methods remove discontinuous depths and interpolate to nearby depths [24,25,26]. It is effective in removing noise in low-texture areas, but sometimes, the boundaries of objects become unclear, and edge bleeding artifacts cannot be removed. The FM operator’s adaptive window method was proposed to compensate for this disadvantage [27]. A large window is used in the low-texture area, and a small window is used at the edge of the object to make the boundary of the object clear. However, the edge bleeding artifact cannot be removed because it is not related to the window size. As another method, a correlation-based guide filtering method was proposed [28]. The boundary of the object is made clear by estimating a depth map with a high correlation between the image and FM value. This method also does not significantly remove edge bleeding artifacts.

In some applications, such as bonding wire inspection, edge bleeding artifacts affect the inspection adversely, so there is a need to eliminate them as much as possible. Therefore, this study analyzed the edge bleeding artifact, which was only known empirically, using a formula based on the gradient at the edges. The edge bleeding artifacts were analyzed about the cause of the artifact and the factors affecting the artifact with microscope images. Various FM operators are compared and verified through simulations because the analysis may vary depending on the type of FM operator. Furthermore, based on the analyzed content, this paper proposes an edge bleeding artifact reduction method to alleviate bleeding artifacts. Lastly, this study compared and verified the performance of the proposed edge bleeding artifact reduction method using microscopy images.

## 2. Analysis of Edge Bleeding Artifact

### 2.1. Principles of Blur Effect by Defocus

General optical principles of SFF are introduced before analyzing edge bleeding artifacts. SFF estimates the depth using focal information. Focal information indicates the degree of focus and defocus of the image. Focal information cannot be obtained if the image does not defocus when the focal distance changes. Therefore, the focusing range must be narrowed to increase the depth resolution. The allowable focal range of the camera is the depth of field (DoF) expressed as Equation (1) [29].
(1)$$DoF\approx \frac{2Ncd^{2}}{f^{2}}$$
where f is the focal length; d is the focal distance, which is the distance to the focus position; and c is the circle of confusion. N is the f-number, N=f/A, and a parameter that is proportional to the DoF. A is the aperture diameter. Equation development was simplified by approximating the DoF using the thin-lens formula [30]. The effect of the defocus blur occurring at the object edge, except for the effect by the object texture, was checked by placing objects 1 and 2 at object distance $u_{1}$ and $u_{2}$, respectively, as shown in Figure 2. Assume that the image of object 1 in focus is a single-tone image $I_{1}\left(x,y\right)=b_{1}$ and the image of object 2 in focus is a single-tone image $I_{2}\left(x,y\right)=b_{2}u\left(x\right)$. Here, $u\left(x\right)$ is the unit step function, and domain x and y are coordinates in the image sensor plane. Because object 1 is masked by object 2, the defocus image $I_{d}$ at focal distance d taken by applying the transmittance map, $T\left(x,y\right)=u\left(-x\right)$, of the object 2 layer, as shown in Equation (2).
(2)$$I_{d}=\left(I_{1}*h_{u_{1},d}\right)\circ \left(T*h_{u_{2},d}\right)+I_{2}*h_{u_{2},d}$$
where $h_{u,d}$ is the point spread function (PSF) at focal distance d. The defocused image is obtained by convolution of the object image and the PSF corresponding to object distance u. The defocused image $\left(I_{1}*h_{u_{1},d}\right)$ of object 1 is masked by element-wise product (∘) with the defocused transmittance map $\left(T*h_{u_{2},d}\right)$ of object 2. The PSF depends on the structure of the camera lens and the shape of the aperture but can be approximated in the form of a Gaussian function [31].
(3)$$h_{u,d}\left(x,y\right)=\frac{1}{2\pi \sigma_{u}^{2}\left(d\right)}\exp\left(-\frac{x^{2}+y^{2}}{2\sigma_{u}^{2}\left(d\right)}\right)$$
where $\sigma_{u}\left(d\right)$ is the blur parameter of the PSF, which is simplified by the object distance and camera parameters when applying the thin-lens formula, as expressed in Equation (4) [32].
(4)$$\sigma_{u}\left(d\right)\approx \frac{f^{2}\left|d-u\right|}{2Nu\left(d-f\right)}$$

For the focus plane to exist, the focal distance d must satisfy the condition d>f. The blur parameter $\sigma_{u}\left(d\right)$ is inversely proportional to the f-number N. Therefore, the blur parameter $\sigma_{u}\left(d\right)$ increases when the f-number N is reduced to shorten the DoF, and the blur caused by defocus becomes stronger.

### 2.2. Occurrence Conditions for Edge Bleeding Artifact

This study analyzed the conditions for the occurrence of edge bleeding artifacts. To analyze why edge bleeding artifact occurs, it is necessary to know the maximum value of the image plane gradient according to the focal distance in the image sequence in the near edge area and the corresponding focal distance. First, the image gradient $\left\|\nabla I_{d}\right\|$ is calculated by defocus using the Euclidean norm of the $I_{d}$ gradient, as shown in Equation (5).
(5)$$\left\|\nabla I_{d}\left(x,y\right)\right\|=\sqrt{{\left(\frac{\partial }{\partial x}I_{d}\right)}^{2}+{\left(\frac{\partial }{\partial y}I_{d}\right)}^{2}}=\frac{\left|b_{2}-b_{1}\right|}{\sqrt{2\pi \sigma_{u_{2}}^{2}\left(d\right)}}\exp\left(-\frac{x^{2}}{2\sigma_{u_{2}}^{2}\left(d\right)}\right)$$

The gradient increases close to the edge and when the difference in brightness between two objects is large. The depth estimation method in SFF finds the focus distance when the FM value is a maximum. Therefore, the gradient according to the focus distance is important. Figure 3a shows the graph to see the change according to blur parameter $\sigma_{u_{2}}\left(d\right)$ of the gradient $\left\|\nabla I_{d}\right\|$. The gradient $\left\|\nabla I_{d}\right\|$ had a single peak and gradually decreased after passing the peak. In addition, the value and position of the peak of the gradient changed when the distance x from the edge changed. This maximum value and position were calculated by a partial derivative of the gradient $\left\|\nabla I_{d}\left(x,y\right)\right\|$ using the parameter $\sigma_{u_{2}}$.
$$\frac{d}{d\sigma_{u_{2}}}\left\|\nabla I_{d}\left(x,y\right)\right\|=\left(\frac{x^{2}}{\sigma_{u_{2}}^{2}\left(d\right)}-1\right)\frac{\left|b_{2}-b_{1}\right|}{\sqrt{2\pi }\sigma_{u_{2}}^{2}\left(d\right)}\;\exp\left(-\frac{x^{2}}{2\sigma_{u_{2}}^{2}\left(d\right)}\right)=0$$
(6)$$∴\sigma_{u_{2}}\left(d\right)=0,\;\left|x\right|$$

Here, $\left\|\nabla I_{d}\left(x,y\right)\right\|$ has a maximum value, as shown in Equation (7), when $\sigma_{u_{2}}\left(d\right)=\left|x\right|$.
(7)$$\underset{\sigma_{u_{2}}}{\max}\left\|\nabla I_{d}\left(x,y\right)\right\|=\frac{\left|b_{2}-b_{1}\right|}{\sqrt{2\pi }}e^{-\frac{1}{2}}\frac{1}{\left|x\right|}$$

Ignoring the texture of the object, a formula was developed to determine the maximum value and position of the gradient because of the differences in brightness. Object 2 was in focus when $\sigma_{u_{2}}\left(d\right)=0$ and $\sigma_{u_{2}}\left(d\right)$ increased as the focal distance changed, according to Equation (4). The maximum gradient decreased gradually in proportion to $1/\left|x\right|$. In addition, the focal distance when the gradient was maximum at position x was the depth $u_{a}$ of the edge bleeding artifact and was calculated as an inverse function of Equation (4).
(8)$$u_{a}^{+}=\frac{Nu_{2}fx+f^{2}u_{2}}{Nu_{2}x+f^{2}},\;u_{a}^{-}=\frac{Nu_{2}fx-f^{2}u_{2}}{Nu_{2}x-f^{2}}$$

$u_{a}$ was calculated with two solutions except for x=0, as shown in Figure 3b.

In general, the FM operator detected the edge blur component and the object texture component together because the SFF is performed in an environment with the object texture. Equation (9) expresses the FM value $v_{d}$ calculated using the FM operator for the image $I_{d}$ taken at the focal distance d.
(9)$$v_{d}=FMO\left(I_{d}\right)$$
where $FMO\left(I\right)$ is a function of the FM operator, which outputs the FM value when image I is input. The depth was estimated as the distance at which the FM value has the maximum value for the x, y points of the image. Therefore, the conditions under which edge bleeding artifacts occur are as follows.
(10)$$v_{u}\left(x,y\right)<v_{u_{a}}\left(x,y\right)$$

Edge bleeding artifacts occur when the depth $u_{a}$ is estimated instead of object depth u because the FM value $v_{u_{a}}\left(x,y\right)$ at $u_{a}$ is greater than the FM value $v_{u}\left(x,y\right)$.

Figure 4 presents the results of performing SFF in an environment where the texture of object 1 is weak and the texture of object 2 is strong. The edge bleeding artifact occurred near the edge toward object 1, where the texture was weak (Figure 4a). Figure 4b shows the FM values in the three areas marked with squares in Figure 4a. At the object 1 point, the FM value was maximized at depth $u_{1}$, and the object 2 point, the FM value was maximized at depth $u_{2}$, so the depth was correctly estimated. On the other hand, at the near edge point, the FM value was maximized at depth $u_{a}^{-}$ or $u_{a}^{+}$, so the depth was misestimated. Figure 4c presents the depth for the vertical line marked in Figure 4a and clearly shows the edge bleeding artifact.

### 2.3. Factors Causing Edge Bleeding Artifact

The factors that cause edge bleeding artifacts by defocus include the DoF, object texture intensity, the brightness difference between objects, and object slope. First, the DoF affects the blur parameter $\sigma_{u}$ by defocus. The f-number N is reduced when the DoF is shortened, the blur parameter $\sigma_{u}$ is increased, and the area where the edge bleeding artifact occurs is widened.

Second, the object texture intensity affects the FM value $v_{u}\left(x,y\right)$ of the object, as shown in Figure 5a. An edge bleeding artifact occurs when the FM value $v_{u_{a}}\left(x,y\right)$ by edge blur is greater than the FM value $v_{u}\left(x,y\right)$ of the object, as expressed in Equation (10). Therefore, the less the object texture, the smaller the FM value of the object, resulting in more edge bleeding artifacts.

Third, the difference in brightness between the two objects $\left|b_{2}-b_{1}\right|$ is proportional to the maximum value of the gradient $\left\|\nabla I_{d}\left(x,y\right)\right\|$, as expressed in Equation (7). Therefore, the FM value $v_{u_{a}}\left(x,y\right)$ at the near edge area increases as the difference in brightness $\left|b_{2}-b_{1}\right|$ increases, as shown in Figure 5b, resulting in more edge bleeding artifacts.

Finally, when the object is tilted, the distance changes depending on the position, and the blur parameter $\sigma_{u}$ also changes. The blur gradient at the near edge area becomes stronger when the object is tilted in such a way that the object distance increases as it moves away from the object edge. The blur gradient at the near edge area is weakened when the slope is reversed. Therefore, the FM value at the near edge tilted to one side, resulting in a unidirectional edge bleeding artifact, as shown in Figure 5c.

## 3. Method to Reduce Edge Bleeding Artifact

The FM value in the near edge area includes the gradient component by defocus blur. As a result of the analysis in Section 2.2, an edge bleeding artifact occurs when the FM value of the gradient component becomes larger than that of the object texture component. Therefore, this paper proposes a method that reduces the edge bleeding artifacts by reducing the FM value of the defocus blur component. The image brightness changes rapidly in the depth of the edge bleeding artifact because the edges are blurred by defocus. The image brightness calculated from Equation (2) is as follows.
(11)$$I_{d}\left(x,y\right)=\frac{\left(b_{2}+b_{1}\right)}{2}+\frac{\left(b_{2}-b_{1}\right)}{2}\mathrm{erf}\left(\frac{x}{2\sigma_{u_{2}}^{2}\left(d\right)}\right)$$
where $\mathrm{erf}\left(x\right)$ is the Gaussian error function and is expressed as $\mathrm{erf}\left(x\right)=2/\sqrt{\pi }\int_{0}^{x}e^{-k^{2}}dk$. Image brightness according to depth at a position near the edge is shown in Figure 6a. The local brightness was the same as the brightness of the object when in focus in the near edge area. On the other hand, when defocused, the local brightness was intermediate between the brightness of the two objects because of the blur effect. The local brightness was asymmetric based on the depth at which the edge bleeding artifact occurred. Nevertheless, the local average brightness between in-focus and defocus did not change significantly at locations other than the edge. In addition, the local brightness is symmetrical based on the in-focus because defocus in the direction close to the lens and defocus in the direction far away from the focal distance have a similar degree of blur effect. Therefore, the edge bleeding artifact reduction method uses measurements of the symmetry and asymmetry of the local brightness.

The window size for calculating local average brightness uses the same size as the FM operator. The equation for calculating the local average brightness $B_{d}$ in the image sequence $I_{d}$ is as follows.
(12)$$B_{d}\left(x,y\right)=\frac{1}{{\left(2k+1\right)}^{2}}\sum_{i=-k}^{k}\sum_{j=-k}^{k}I_{d}\left(x+i,y+j\right)$$
where k is half the window size W and is calculated as $k=\left(W-1\right)/2$. The symmetry and asymmetry of the reference point $B_{d}$ in the depth domain were calculated. For the offset invariant, subtracting the offset from each element of the reference point $B_{d}$ gives $B_{d+i}^{*}=B_{d+i}-B_{d}$. Here, the even component is $B_{d+i}^{*}+B_{d-i}^{*}=B_{d+i}+B_{d-i}-2B_{d}$, and the odd component is $B_{d+i}^{*}-B_{d-i}^{*}=B_{d+i}-B_{d-i}$. Therefore, the symmetric degree $\alpha_{d}\left(x,y\right)$ and asymmetric degree $\beta_{d}\left(x,y\right)$ in the range [−γ,γ] were calculated using Equation (13).
$$\alpha_{d}\left(x,y\right)=\sqrt{\sum_{i=1}^{\gamma }{\left(B_{d+i}\left(x,y\right)+B_{d-i}\left(x,y\right)-2B_{d}\left(x,y\right)\right)}^{2}}$$
(13)$$\beta_{d}\left(x,y\right)=\sqrt{\sum_{i=1}^{\gamma }{\left(B_{d+i}\left(x,y\right)-B_{d-i}\left(x,y\right)\right)}^{2}}$$

The local average brightness is asymmetric where the edge bleeding artifact occurs and symmetric at the in-focus point on the object. Therefore, the FM value of the points where the local average brightness is asymmetric should be reduced to reduce the edge bleeding artifact. FM value weight for reducing edge bleeding artifacts is as follows.
(14)$$w_{d}\left(x,y\right)=\frac{\alpha_{d}\left(x,y\right)+\eta \gamma }{\alpha_{d}\left(x,y\right)+\eta \gamma +\beta_{d}\left(x,y\right)}$$
where η is the symmetry sensitivity coefficient and is adjusted so that $w_{d}\left(x,y\right)$ is calculated as symmetry when the local average brightness is flat. The range of $w_{d}\left(x,y\right)$ is [0, 1]. In the case of complete even symmetry, $w_{d}\left(x,y\right)=1$, and in the case of asymmetry, $w_{d}\left(x,y\right)$ is close to zero. The weighted FM value $v_{d}^{weighted}\left(x,y\right)$ is calculated by multiplying the FM value $v_{d}\left(x,y\right)$ obtained with the FM operator by the edge bleeding artifact reduction weight $w_{d}\left(x,y\right)$.
(15)$$v_{d}^{weighted}\left(x,y\right)=w_{d}\left(x,y\right)v_{d}\left(x,y\right)$$

Figure 6b shows the weighted FM value. Finally, the depth map $\hat{D}\left(x,y\right)$ is estimated as the index with the maximum weighted FM value.
(16)$$\hat{D}\left(x,y\right)=arg\underset{d}{\max}\left(v_{d}^{weighted}\left(x,y\right)\right)$$

## 4. Results

### 4.1. Simulation

Simulations were performed to verify the method analyzed and proposed in this paper. First, the degree of edge bleeding artifact was compared according to the FM operator type. The factors affecting the edge bleeding artifact were then verified. Lastly, the proposed edge bleeding artifact reduction method was verified.

The FM operators to be compared and verified are the gradient-based operator (GRA) [16], Laplacian-based operator (LAP) [17], statistic-based operator (STA) [20], and Wavelet-based operator (WAV) [18]. Table 1 lists the simulation parameters. The size of the object and the real image are the same because the focal distance is twice the focal length. Therefore, one image pixel corresponds to 1 μm of the object plane because the image sensor cell size is 1 μm. In addition, the calculated DoF was 20 μm when the circle of confusion value was substituted into the image sensor cell size.

The evaluation measurements for comparing the performance of simulation results include the root mean square error (RMSE) and the width of the edge bleeding artifact. The RMSE measures the error between the ground truth and the estimated depth map. The RMSE measurement was performed around the edge as a region of interest to increase the impact of the edge bleeding artifact. When the size of the region of interest is W×H, the ground truth is $D_{T}\left(x,y\right)$, and the estimated depth map is $\hat{D}\left(x,y\right)$, the RMSE is as follows.
(17)$$RMSE=\sqrt{\frac{1}{WH}\sum_{\left(x,y\right)}{\left(D_{T}\left(x,y\right)-\hat{D}\left(x,y\right)\right)}^{2}}$$

The edge bleeding artifact occurs when the RSME is larger. On the other hand, the width of the edge bleeding artifact was also measured as a measurement independent of the depth difference because the RMSE is dependent on the depth difference between the two objects. The width of the edge bleeding artifact is the average width of the edge bleeding artifact in the direction perpendicular to the object edge. The average width was calculated by excluding the top and bottom 20% to remove outlier data.

Figure 7 shows the edge bleeding artifact according to FM operators. The area corresponding to the red box shown in Figure 7a was enlarged in all result figures to check edge bleeding artifacts. Results were expressed in the same color scale. The edge bleeding artifact occurs in all types of FM operators, even though there are differences in the degree. Table 2 presents a quantitative evaluation of the performance. Blurring affects the gradient and statistic-based FM operators, resulting in many edge bleeding artifacts. Wavelet-based FM operator detects high-frequency components and is indirectly affected by blur, resulting in weak edge bleeding artifacts. The Laplacian-based FM operator is the second derivative and is most robust against edge bleeding artifacts as it is affected by the amount of gradient change caused by blur.

Figure 8 compares the edge bleeding artifacts according to the DoF, texture, and brightness differences between objects. The DoF was adjusted by modifying the f-number, and the texture was adjusted by changing the contrast of object 1. The texture strength ratio is the ratio of the contrast of object 2 and that of object 1. Figure 8a compares the DoF. As the DoF becomes shorter, the blur widens, and the width of the bleeding artifact widens. The longer DoF reduces edge bleeding artifacts but reduces depth resolution and increases noise. Therefore, RMSE decreases as the DoF lengthens as the edge bleeding artifact decreases but does not decrease continuously due to noise. Figure 8b compares the edge bleeding artifacts according to texture. As the texture weakens, the edge bleeding artifact becomes stronger in all FM operators. Figure 8c shows the differences in brightness between the objects. The edge bleeding artifact becomes stronger in all FM operators as the brightness difference increases.

Figure 9 shows the edge bleeding artifact according to the slope of object 2. Edge bleeding artifacts were compared for the following: a positive slope, where the object distance decreases as the distance from the edge increases; a negative slope, where the object distance increases; and a flat, where the object distance is constant. When the object was tilted, the defocus gradient in one direction became stronger, and the defocus gradient in the other direction became weaker. Therefore, in the case of flat, the edge bleeding artifact was a mixture of cases where depth increases and decreases. On the other hand, with a positive slope, only edge bleeding artifacts occurred in the direction that the depth decreases. In contrast, with negative slopes, only edge bleeding artifacts occurred in the direction that the depth increased.

Figure 10 shows the results of the proposed edge bleeding artifact reduction method. Table 3 compares the performance of the proposed method. When the proposed method was applied, the edge bleeding artifact was reduced in all FM operators. In particular, this shows excellent performance in Gradient-based and Wavelet-based FM operators. RMSE and width decreased by 68.0% and 83.2% in GRA and by 65.8% and 85.3% in WAV. In STA, it decreased by 39.5% and 50.9%, but edge bleeding artifacts still remain. Compared to other FMOs, STA has a larger difference between the FM value of the object and the FM value of the edge. Even if the proposed method was applied, the FM value of the edge was not sufficiently reduced, so the reduction rate of the edge bleeding artifact was low. LAP had the lowest reduction rate, but there was almost no edge bleeding artifact, even in the initial depth map. Because each FM operator has a different sensitivity to object texture and edge blur, the performance of the proposed method differs for each FM operator.

### 4.2. Implementation in Microscopic System

An experimental environment was configured to verify the edge bleeding artifact reduction method proposed in this paper in a microscope environment, as shown in Figure 11. The microscope lens used was SZ6CHIF3-B5 from Sunny Optical. The DoF was 20 µm, and the lateral resolution was approximately 1 µm. The camera used was Teledyne DALSA’s G3-GM10-M1280. The image resolution was 1280 × 1024, and the sensor size was 1/2 inch. A method of adjusting the actual distance between the object and the lens was adopted without using a focal distance adjustment ring to ensure linearity of depth according to the image sequence. The stepper motor moves the microscope vertically by moving the screw shaft. The moving distance per step was 2.5 µm. A gradient-based operator was used as the FM operator, and the window size was 21. The compared methods are adaptive window size [27] and guided filtering [28]. The proposed method was used together with a postprocessing algorithm. The resulting quality of the compared method and the proposed method may differ. Therefore, when looking at the results, attention should be paid to the edge bleeding artifact, which is the subject of this study.

Figure 12 presents the result of 3D imaging of the bonding wire connecting the semiconductor and lead of the IC. The above line figures are the all-in-focus image and the result of applying each FM operator, and the bottom line figures are enlarged images of the red boxes. Figure 12a,b show a focus image and initial depth map performed by SFF using a gradient-based FM operator. The width of the edge bleeding artifact in the initial map is 58.1 µm. Figure 12c shows the results of the adaptive window size method, and the width of the artifact is measured to be 40.1 µm, which has a 30.9% reduction performance. Figure 12d shows the results of the guided filtering method, and the width of the artifact is measured to be 37.8 µm, resulting in a 34.9% reduction performance. Figure 12e shows the results of the proposed method, and the width of the artifact is measured to be 10.3 µm, resulting in a reduction performance of 82.2%. A closer look at the all-in-focus image showed that the semiconductor, bottom layer, and bonding wire have strong textures. On the other hand, the shaded area of the bottom layer had a weak texture. As a result of SFF, the depth of the wire bonded to the semiconductor was calculated. Nevertheless, noise occurred due to an incorrect depth estimation in some low-contrast areas. Furthermore, edge bleeding artifacts occurred in locations where the brightness difference between the objects was large, and the texture was weak. The positions marked with red squares were enlarged and shown below in the original results image. Many edge bleeding artifacts occurred between the semiconductor and the bottom layer. In addition, an edge bleeding artifact occurred between the wire and the bottom layer. The noise was reduced in the results of the adaptive window size method and guided filtering method, but edge bleeding artifacts remained. The edge bleeding artifact was greatly reduced after the proposed edge bleeding artifact reduction method.

Figure 13 presents the result of 3D imaging of nylon thread. The above line figures are the all-in-focus image and the result of applying each FM operator, and the bottom line figures are enlarged images of the red boxes. The width of the measured edge bleeding artifact is 42.9 µm for the initial method, 33. 2 µm for the adaptive window method, 32.1 µm for the guided filtering method, and 12.9 µm for the proposed method. The reduction percentage of the compared methods is about 22%, while the reduction percentage of the proposed method is 70%. In the focus image, Figure 13a, the bottom layer has a strong texture, but some areas of the nylon thread have a weak texture. Furthermore, there was a large difference in brightness between the nylon thread and the bottom layer. In this case, edge bleeding artifacts occurred in the direction of the nylon thread with a weak texture. Figure 13b shows that considerable noise occurred in the weak-texture area because SFF was performed using the gradient-based FM operator. Figure 13c,d show that some noise was removed in compared methods, but edge bleeding artifacts remained. Figure 13e shows that the edge bleeding artifact was reduced noticeably after applying the edge bleeding artifact reduction method.

## 5. Conclusions

Edge bleeding artifacts, where depths near the object edge are bleeding, frequently occur when performing SFF on microscopic images. This study analyzed the causes and influencing factors of edge bleeding artifacts. The gradient component due to the difference in brightness between objects during defocus was calculated after excluding the effect of the object texture. As a result of the calculation, an edge bleeding artifact occurred when the FM value of the defocus gradient was greater than that of the object texture. Furthermore, factors that cause edge bleeding artifacts include the camera’s short DoF, weak object texture, large brightness difference between two objects, and object tilt. The analysis results were verified through simulation. In addition, the degree of the edge bleeding artifact was evaluated in various FM operators.

Second, this paper proposed a method that alleviates edge bleeding artifacts based on the previous analysis. Local brightness is symmetry based on the in-focus depth and asymmetry based on the depth where edge bleeding artifacts occur. Therefore, the weight of the FM value was calculated using the symmetry and asymmetry of local brightness. The proposed method reduces the FM value at the depth where edge bleeding artifacts occur. Therefore, the FM value of the object texture becomes larger than the FM value of the defocus gradient, thereby relieving the edge bleeding artifacts. The proposed method was verified through simulation and implementation. As a result of the simulation, the edge bleeding artifacts were reduced regardless of the type of FM operator. In particular, RMSE decreased by more than 60% in the gradient-based FM operator and wavelet-based FM operator. In the implementation, a qualitative evaluation was performed because there was no ground truth information. The analysis showed that edge bleeding artifacts occurred in areas where the object texture was weak and the brightness difference was large. A significant amount of edge bleeding artifacts remained even when applying the existing compared algorithms, but the edge bleeding artifact was reduced after applying the proposed method.

A limitation of the proposed method is that the depth estimation accuracy in the weak-texture areas was lowered. The FM value decreased slightly because the local brightness was not perfectly symmetrical based on the focus. This phenomenon does not affect the depth estimation in general texture areas but causes an inaccurate depth estimation in weak-texture areas because of the slight decrease in FM value. Therefore, the depth estimation noise increased when the proposed method was used alone. Hence, it is recommended to use it together with a postprocessing algorithm.

The depth estimation problem in low-contrast and weak-texture areas is a challenge in most SFF-related papers. The method proposed in this paper also showed poor performance in the weak-texture area. On the other hand, the deterioration of the depth estimation performance can be prevented if the proposed method is combined with information, such as reliability [33] or FM value entropy [24]. Future work will examine ways of improving the edge bleeding artifact reduction method using weak-texture area estimations to overcome the limitations of the proposed method.

## Figures and Tables

**Figure 1 sensors-23-08602-f001:**
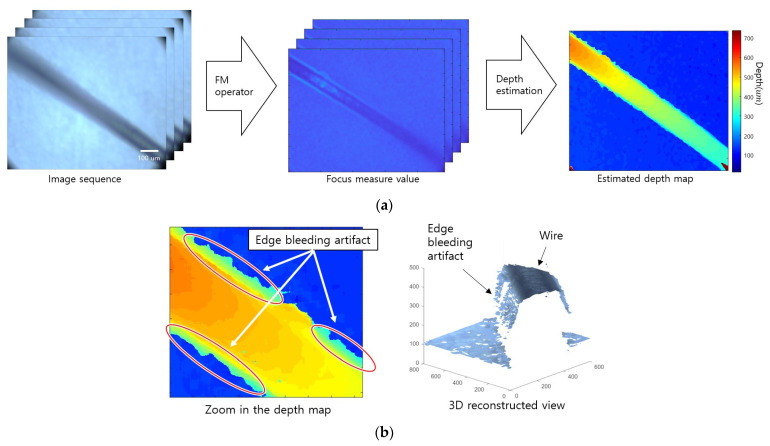
Example of shape from focus and edge bleeding artifact: (**a**) traditional method of shape from focus and (**b**) examples of edge bleeding artifact. The estimated depth is bleeding at the wire edge.

**Figure 2 sensors-23-08602-f002:**
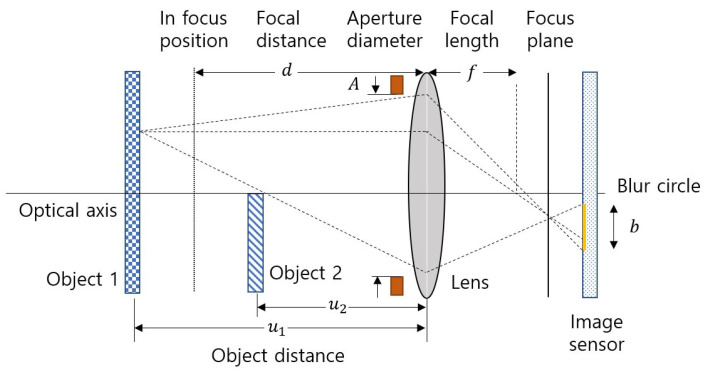
Defocus formation using the lens model.

**Figure 3 sensors-23-08602-f003:**
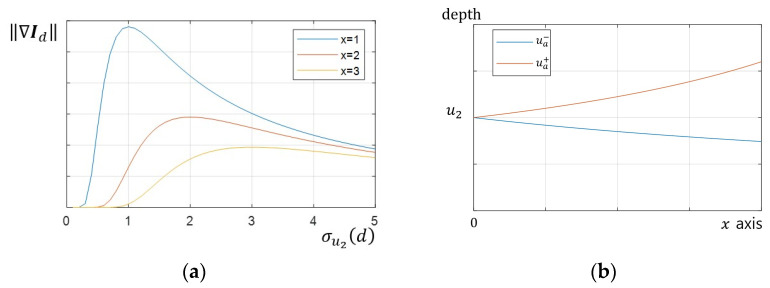
Tendency of variables: (**a**) blur parameter vs. gradient graph, gradient has a single maximum; (**b**) depth of edge bleeding artifact depending on position.

**Figure 4 sensors-23-08602-f004:**
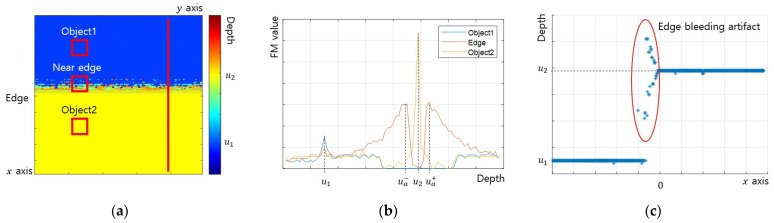
Example of SFF results: (**a**) estimated depth map; (**b**) FM value at points; (**c**) estimated depth along *x* axis.

**Figure 5 sensors-23-08602-f005:**
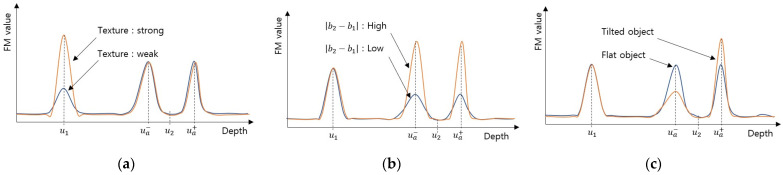
The FM value according to factors causing edge bleeding artifact: (**a**) comparison according to object texture; (**b**) comparison according to $\left|b_{2}-b_{1}\right|$; (**c**) comparison according to object slope.

**Figure 6 sensors-23-08602-f006:**
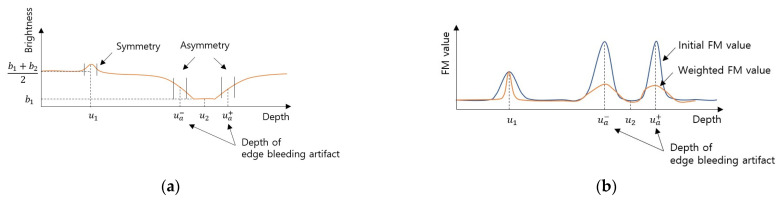
Local brightness and FM value near the edge: (**a**) Depth vs. local brightness. Asymmetry in the depth of the edge bleeding artifacts. (**b**) Comparison of initial FM value and weighted FM value.

**Figure 7 sensors-23-08602-f007:**
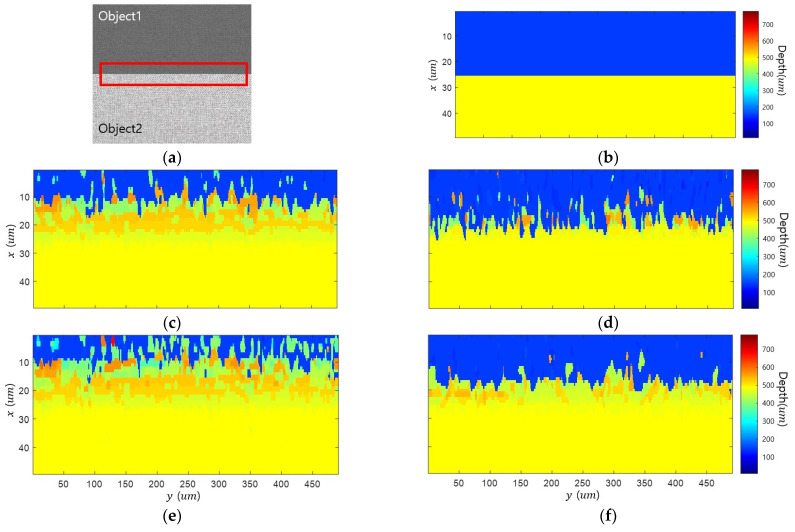
Simulation results of various FM operators: (**a**) all-in-focus image; (**b**) ground truth; (**c**) gradient-based FM operator; (**d**) Laplacian-based FM operator; (**e**) statistic-based FM operator; (**f**) wavelet-based FM operator.

**Figure 8 sensors-23-08602-f008:**
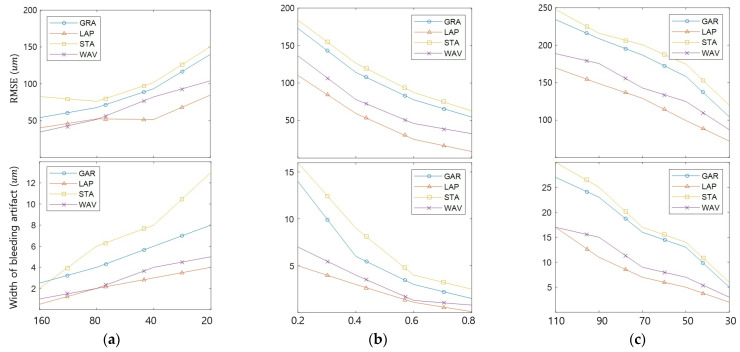
Comparison of edge bleeding artifact occurrence tendencies according to factors: (**a**) lens DoF; (**b**) object texture strength; (**c**) brightness difference between objects.

**Figure 9 sensors-23-08602-f009:**
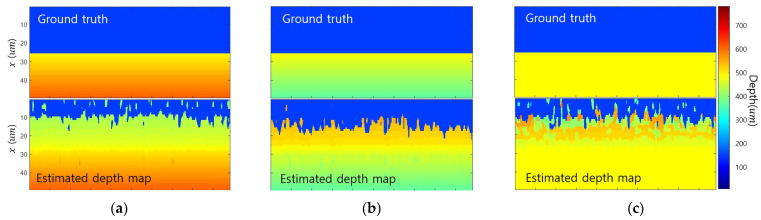
Comparison of edge bleeding artifact according to the slope of object: (**a**) positive slope; (**b**) negative slope; (**c**) flat.

**Figure 10 sensors-23-08602-f010:**
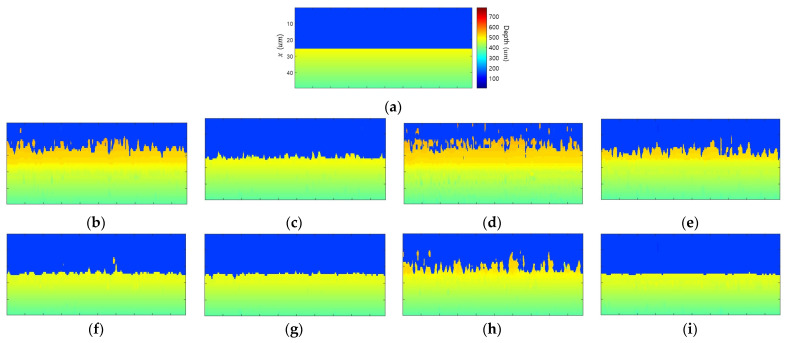
Simulation results of proposed edge bleeding artifact reduction method: (**a**) ground truth; (**b**–**e**) initial depth map of (**b**) GRA, (**c)** LAP, (**d**) STA, and (**e**) WAV; (**f**–**i**) depth map estimated with the proposed weighted FM value of each FM operator.

**Figure 11 sensors-23-08602-f011:**
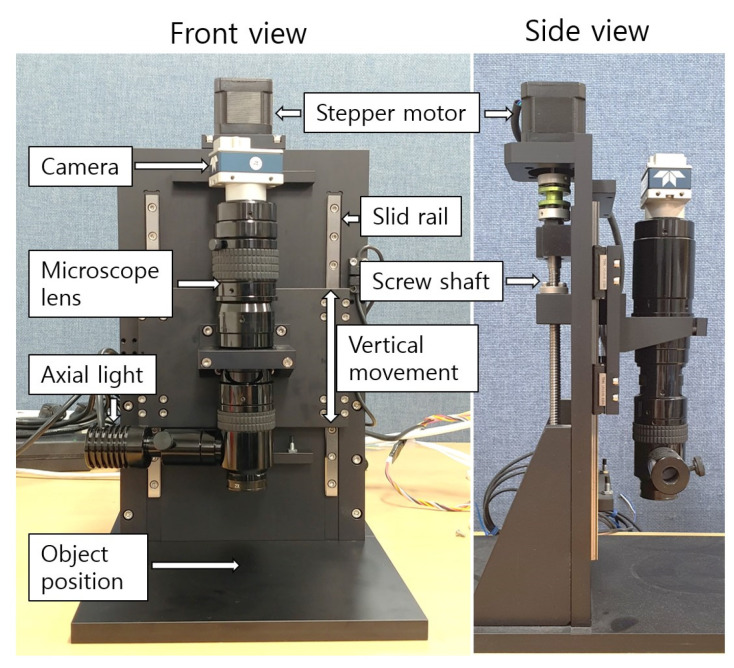
Microscopic device used in the experiment.

**Figure 12 sensors-23-08602-f012:**
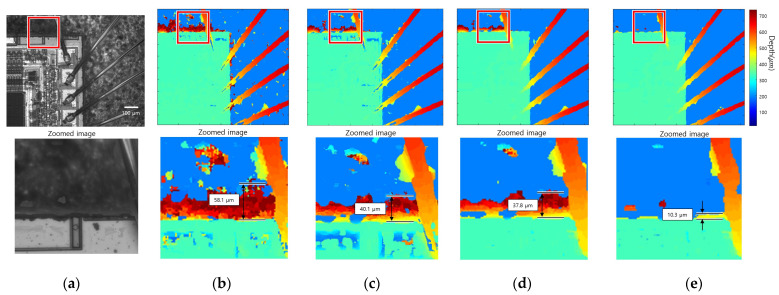
Bonding wire connecting the semiconductor and lead of the IC: (**a**) all-in-focus image; (**b**) initial depth map; (**c**) adaptive window size method; (**d**) guided filtering method; (**e**) proposed method.

**Figure 13 sensors-23-08602-f013:**
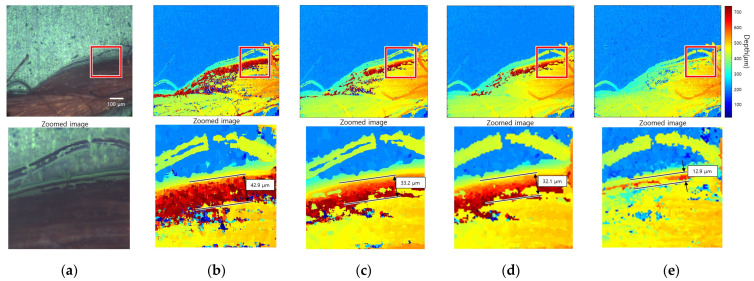
Nylon thread: (**a**) all-in-focus image; (**b**) initial depth map; (**c**) adaptive window size method; (**d**) guided filtering method; (**e**) proposed method.

**Table 1 sensors-23-08602-t001:** Simulation parameter.

Symbol	Description	Value	Unit
	Image size	500 × 500	pixel
c	Circle of confusion	1	μm
f	Focal length	5	μm
d	Focal distance	10	mm
N	f-number	2.5	

**Table 2 sensors-23-08602-t002:** Evaluations of various FM operators.

Evaluation Measurement	GRA	LAP	STA	WAV
RMSE (μm)	186.2	129.6	201.4	142.1
Width of artifact (μm)	16.3	7.1	17.9	9.5

**Table 3 sensors-23-08602-t003:** Performance evaluations of the proposed edge bleeding artifact reduction method.

FM Operator	Method	RMSE (µm)	Reduction Percentage	Width of Artifact (µm)	Reduction Percentage
GRA	Initial	169.7	68.0%	14.3	83.2%
	Proposed method	54.4		2.4	
LAP	Initial	61.9	21.3%	2.7	40.7%
	Proposed method	48.1		1.6	
STA	Initial	187.5	39.5%	16.1	50.9%
	Proposed method	113.4		7.9	
WAV	Initial	123.7	65.8%	7.5	85.3%
	Proposed method	42.3		1.1	

## Data Availability

Not applicable.

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
