# Peer review of "Edge Bleeding Artifact Reduction for Shape from Focus in Microscopic 3D Sensing"

_sensors, 2023, doi:10.3390/s23208602_

Round 1

Reviewer 1 Report

The submitted manuscript presents the improvement of shape from focus methods by reducing the edge bleeding artifacts by reducing the FM (focus measure) value of the defocus blur component. The proposed method is verified with examples showing its effectiveness on four different types of FM operators, reducing the RMSE by more than 60% and the width of the artifacts by more than 80% in the case of the gradient-based (GRA) and wavelet-based (WAV) FM operators. The manuscript is generally well written, with a clearly explained method for reducing edge bleeding artifacts and clearly presented results. Of course, there are some minor issues whose elimination would improve the quality of the work.

First, regarding the English language, there are some errors that could be easily spotted and corrected by careful reading. For example, on page 3, line 89, part of a sentence is omitted. 

Second, it seems to me that there is a slight problem with the organization of the material in the manuscript, particularly in Section 2, in that there is no clear distinction between what is adopted and what is a contribution. Since the equations are referenced in subsection 2.1, this subsection provides a general introduction to the problem of focusing and determining the blur parameter. Subsections 2.2 and 2.3 contain figures and equations that are not referenced. In addition, there is no reference in these subsections, which should indicate that this is a contribution to ongoing research in the area of detection of edge bleeding artifacts, but this is not emphasized anywhere. The name of the function (FMO) in Equation 9 should be associated with the "FM operator" that appears in the previous text. It can be inferred from the context, but should also be explicitly stated, since it is not a standard function.

What does "k" mean in the equation in line 207 and how is it related to the window size (2k+1) in equation 12?

With respect to the simulation results shown in Figure 10 and Table 3, it would be good to explain why the proposed method did not remove the edge bleeding artifacts for the STA FM operator. It would be good to explain how the proposed method affects the different FM operators and what the reduction efficiency depends on.

Section 3 was incorrectly referred to as Section 2.

The English is clear and understandable, but there are some errors that can be easily spotted and corrected by careful reading. For example, on page 3, line 89, part of a sentence is omitted.

Author Response

Thank you very much for taking the time to review our paper. We are sincerely grateful for your valuable feedback and insightful comments on our work. Below are our answers to your comments.

  1. (comment) First, regarding the English language, there are some errors that could be easily spotted and corrected by careful reading. For example, on page 3, line 89, part of a sentence is omitted. 

-> The sentence has been modified as follows: “The edge bleeding artifacts were analyzed about the cause of the artifact and the factors affecting the artifact with microscope images.” We also checked the manuscript as a whole and corrected any problematic areas.

  1. (comment) Second, it seems to me that there is a slight problem with the organization of the material in the manuscript, particularly in Section 2, in that there is no clear distinction between what is adopted and what is a contribution. Since the equations are referenced in subsection 2.1, this subsection provides a general introduction to the problem of focusing and determining the blur parameter. Subsections 2.2 and 2.3 contain figures and equations that are not referenced. In addition, there is no reference in these subsections, which should indicate that this is a contribution to ongoing research in the area of detection of edge bleeding artifacts, but this is not emphasized anywhere.

-> Regarding your comment, we agree with your advices and have made the appropriate revisions in the revised manuscript. Section 2.1 explains the general SFF optical principles and blur. The equation was cited and used as the basis for analysis of edge bleeding artifacts. Sections 2.2 and 2.3 are analyzes of edge bleeding artifacts and are a contribution of this study. To clearly distinguish between citations and contributions, emphasis was added to the first sentence of each section.

  1. (comment) The name of the function (FMO) in Equation 9 should be associated with the "FM operator" that appears in the previous text. It can be inferred from the context, but should also be explicitly stated, since it is not a standard function.

-> Added explicit description of FMO function.

  1. (comment) What does "k" mean in the equation in line 207 and how is it related to the window size (2k+1) in equation 12?

-> The window size W for calculating local average brightness uses the same size as the FM operator. k is half the window size and is calculated as k=(W-1)/2. k is used to express equation 12 neatly. Sentences with unclear explanations were deleted and an explanation for k was added.

  1. (comment) With respect to the simulation results shown in Figure 10 and Table 3, it would be good to explain why the proposed method did not remove the edge bleeding artifacts for the STA FM operator. It would be good to explain how the proposed method affects the different FM operators and what the reduction efficiency depends on.

-> The following phrase was added to explain the reason for the difference in performance depending on the FM operator and the factors affecting the performance of the proposed method. ”Compared to other FMOs, STA has a larger difference between the FM value of the object and the FM value of the edge. Even if the proposed method was applied, the FM value of the edge was not sufficiently reduced, so the reduction rate of edge bleeding artifact was low. LAP had the lowest reduction rate, but there was almost no edge bleeding artifact even in the initial depth map. Because each FM operator has different sensitivity to object texture and edge blur, the performance of the proposed method differs for each FM operator.”

  1. (comment) Section 3 was incorrectly referred to as Section 2.

-> Modified to section 3.

Once again, thank you for your thoughtful review. We believe your input has significantly strengthened our paper. Your time and expertise are greatly appreciated.

Reviewer 2 Report

===== Synopsis:

The study proposes an algorithm to sharpen the boundaries of depth images, which typically suffer from edge bleeding. It is tested in simulation and in implementation.

===== General Comments:

The results look quite promising in simulation but it lacks a quantification of the results for the implementation - only a visual comparison is given. The authors say that there exists no annotated data, but couldn't one generate that for a small dataset? It would be more convincing.

I'm afraid I cannot give more intelligent comments. Reading the abstract, I had expected the study to be closer to my work, but this is not exactly my domain.

===== Specific Comments:

Lines 88/89: the word 'why' is confusing.

I understood most of it.

Author Response

Thank you for your valuable feedback on our paper. We appreciate the time and effort you spent in reviewing our work. Below are our answers to your comments.

  1. (comment) The results look quite promising in simulation but it lacks a quantification of the results for the implementation - only a visual comparison is given. The authors say that there exists no annotated data, but couldn't one generate that for a small dataset? It would be more convincing.

-> we agree with your comment about the lack of quantification of implementation results. For improvement, we added the width of the edge bleeding artifact in the depth map of the implementation result. In each experiment result, the width of the edge bleeding artifact was visually expressed and an explanatory sentence about the results was added.

  1. (comment) Lines 88/89: the word 'why' is confusing.

-> The sentence has been modified as follows: “The edge bleeding artifacts were analyzed about the cause of the artifact and the factors affecting the artifact with microscope images.” We also checked the manuscript as a whole and corrected any problematic areas.

Once again, thank you for your thoughtful review.